# Impact of Coherent Laser Irradiation on Germination and Mycoflora of Soybean Seeds—Innovative and Prospective Seed Quality Management

**Agnieszka Klimek-Kopyra** [1,*] **, Joanna Dłużniewska** [2] **, Anna Ślizowska** [1] **and Jan Wincenty Dobrowolski** [3]

[1] Department of Agroecology and Plant Production, University of Agriculture in Kraków, Al. Mickiewicza 21, 31-120 Kraków, Poland; anslizowska@gmail.com

[2] Department of Microbiology and Biomonitoring, University of Agriculture in Kraków, Al. Mickiewicza 21, 31-120 Kraków, Poland; joanna.dluzniewska@urk.edu.pl

[3] Department of Photogrammetry Remote Sensing of Environment and Spatial Engineering, AGH University of Science and Technology Al. Mickiewicza 30, 30-059 Kraków, Poland; dobrowol@agh.edu.pl

* Correspondence: agnieszka.klimek@urk.edu.pl; Tel.: +48-508-984-401

**Abstract:** Laser irradiation is considered a new technology in agriculture; however, the success of irradiation depends on the selection of precise parameters for the light source and exposure. In this study, the impact of laser stimulation on germination and the occurrence of mycoflora in soybean seeds was assessed. The following factors were considered: (1) irradiation using blue and red coherent lights, (2) irradiation of seeds only (a), use of *Bradyrhzobium japonicum* vaccine only (b), and irradiation of the seeds plus the *Bradyrhzobium japonicum* vaccine (c). The germination index, seedling weight and seeds infected by fungus were determined. It was found that the laser treatment of seeds increased germination and seedling weight. Laser irradiation affected the abundance of species *Phoma glomerata*, *Botrytis cinerea*, *Rhizopus nigricans* and *Gliocladium roseum.* The use of blue laser (LB—514 nm) reduced the number of the non-pathogenic species, *R. nigricans* and *G. roseum.*

**Keywords:** sustainable crop production; laser radiation; fungi; soybean

## 1. Introduction

Agricultural sustainability and food security are a priority not only within the European policy framework, but also on a global scale. Interest in the use of lasers in agriculture has existed for some time [1], but in recent years this technology has experienced a real renaissance, indeed, for some it is seen as the only correct way to produce healthy, high-quality food [2]. Laser technology is mainly used for pre-sown seed exposure in order to achieve better and faster germination in various, often unfavourable, habitat conditions. The scientific literature has confirmed that it has various bio-stimulation effects (positive, negative, neutral), which are conditioned by the physical parameters of the laser, including radiation dose and exposure time, and the selection of plant material [2–4]. The inappropriate selection of physical parameters for the plant material may cause unwanted mutagenic effects [5,6]. The ability of a He-Ne laser to repair damage in plants exposed to different intensities of ultraviolet-B radiation was observed in wheat and broad bean cells [7,8]. The authors showed that the properly selected power and time of laser exposure can repair the DNA structure in plant cells and shorten the time of cell regeneration. Podleśna et al. [9] studied the effect of pre-sowing laser radiation of pea seeds on seed biochemical processes, germination rate, seedling emergence, growth rate, and yield. The authors showed that treatment of pre-sown pea seeds with He-Ne laser light increased the concentration of amylolytic enzymes and the content of indole-3-acetic acid (IAA) in pea seeds and seedlings.

The exposure of seeds to He-Ne laser light improved the germination rate and uniformity and modified growth stages, which caused the acceleration of flowering and ripening of pea plants [10]. Research on laser stimulation of soybean seeds is scarce and comes from the 1990s, hence, there was a need to undertake research in the field of laser stimulation of pre-emergent seeds.

Soybean seeds contain fungi on their surface that significantly reduce the germination rate, even after previous treatment [11,12]. Seed-borne diseases are regarded as a major constraint in soybean production. Infected seeds serve as a source for the spread of pathogens in disease-free areas. World-wide, it has been reported that more than 7 Mt of soybean are lost due to disease alone. Some of the seeds are discarded due to low germination, which is influenced by fungal pathogens [13,14]. The species of *Aspergillus, Penicillium, Fusarium, Rhizopus* and *Alternaria* commonly occur as post-harvest molds in storage conditions [15]. The composition of the fungi communities that inhabit soybean seeds may change due to external factors. For example, the fungi observed on soybean seeds harvested during a period of high rainfall in Mato Grosso do Sul, Brazil included *Fusarium* spp. (80–90%), *Phomopsis* spp. (39–45%), *Cercospora* spp. (22–30%), *Colletotrichum* spp. (5–10%), *Rhizoctonia* spp. (<2%) and *Penicillium* spp. [16]. These fungi are responsible for the production of mycotoxins and these metabolites affect the seed quality, germination, viability, seedling vigor, growth of the root and coleoptile [17]. However, information regarding the mycoflora on soybean seeds coupled with laser irradiation is scarce. Wilczek et al. [18] reported that laser light with a surface power density and multiplication of R6x3 and R6x5 completely destroyed *Penicillium* type fungi, whereas in a dose of R3x3, R3x5 and R6x1 it significantly stimulated the growth of *Alternaria* type fungi. Another study showed that hard wheat (*Triticum durum*) seeds irradiated with second harmonic generation (SHG) Nd-YAG laser (532 nm wavelength and total output power of 20 mW, with a beam diameter of 2 mm) were not infected by fungi, while seeds irradiated by two other types of laser (a diode laser (650 nm wavelength with 5 mW output power with beam diameter of 2 mm) and a He-Ne laser ($\lambda$ = 632.8 nm with total maximum output power of 1 mW and beam diameter (2 mm)) showed high rates of infection [19].

Numerous measures have been undertaken to improve soybean germination and its resistance to pathogens, including breeding, and chemical and biological methods, which have not always worked well under the natural conditions of soybean cultivation. The use of laser technology to stimulate seeds and increase the effectiveness of germination was presented in a paper by Ouf and Abdel-Hady [11]. The research was focused on the effect of laser bio-stimulation on the germination capacity of soybean seeds, chlorophyll and carotenoid content and seed health, and found that the effect varied according to the length of exposure. Germination stimulation was noticeable after 1 min of exposure. At this dose, the activity of most seeds increased. Irradiation of soy with laser light for 3 min clearly reduced infection of the seeds with fungi. At the same time, longer exposure inhibited seed germination. The above results indicate that there is no current literature in the field of laser stimulation of pre-sown soybean. The authors showed that the quality of seeds is determined by the length of exposure, however, no studies have been conducted on the effects of short but coherent light stimulation on the health and germination of soybean seeds. In addition, no bacterial vaccine has been considered, which could have a significant effect on the germination and health of exposed seeds. *Bradyrhizobium japonicum* is used for the inoculation of soybean seeds and it is regarded as a "plant growth promoting rhizobacteria" (PGRP). This species of rhizobia has been found to promote plant growth and also to inhibit the growth of various soil-borne pathogens. The growth promotion induced by rhizobia may be directly through nitrogen fixation and the production of plant growth regulators, i.e., substances that stimulate plant growth. the promotion of plant growth may also indirectly suppress fungus pathogens by rhizobia metabolites [20–22]. There are no indigenous *Bradyrhizobium japonicum* strains that form nitrogen-fixing, root nodule symbioses with soybeans in arable lands in Poland, and thus far, only a few local populations have been identified in a few locations where inoculants were applied for growing soybeans [23,24].

　　　The aim of the study was to assess the sowing properties of soybean and the fungal strains that inhabit soybean seeds after irradiating the seeds with combined and short doses of different colors of laser light and depending on the use of a bacterial vaccine.

## 2. Materials and Methods

### 2.1. Experimental Design

　　　Irradiation was carried out at the Department of Environmental Biotechnology and Ecology, AGH University of Science and Technology in Krakow. Certified (uniform) soybean seeds cv. Augusta from the KWS company were used for testing. The selected seed material (100 seeds) was well developed, without discoloration, mycelium deposits and insect or mechanical damage.

　　　The samples were exposed to two types of laser, a Helium-Neon laser (He-Ne, named here as LR) comprised of red light with a wavelength of 632.8 nm and irradiation density of 2 W m$^{-2}$, and an Argon laser (Ar, named here as LB) with a blue light, a wavelength of 514 nm and irradiation density of 5 W m$^{-2}$.

　　　A two factorial experiment was conducted in lab conditions. The first factor was the type of the laser irradiation. The second factor was the irradiated material. The first factor involved four combinations: (1) laser red (LR) was applied three times, for 9 s each time, with a 9 s break between each laser exposure, (2) laser blue (LB) irradiation was applied for 3 × 3 s, with 3 s breaks, (3) LB was executed for 3 × 1 s followed by LR 3 × 3 s, with a 1 s and 3 s break, respectively, and (4) the control sample, without any kind of irradiation. The second factor involved three combinations of irradiated material: (1) irradiation of seeds and vaccine with bacteria *Bradyrhizobium japonicum*, (2) vaccine with bacteria *Bradyrhizobium japonicum* only, and (3) irradiation of the seeds only.

　　　The time duration was controlled by the equipment used in the experiment, according to the methodology of Wilczek et al. [18]. The device is autonomous and includes programmable drivers that can be programmed for specific running times and programmable drivers that activate the appropriate laser at the correct time.

### 2.2. Experimental Procedures

　　　Germination of the seeds was carried out according to the ISTA (International Seed Testing Association) instructions [25] in germination boxes and sterile tissue paper soaked with sterile water served as the substrate. One hundred seeds were tested in five replicates. Germination boxes with the tested seeds were maintained in a growth chamber (Simez Control s.r.o., Vsetin, Czech Republic) for 11 days (days of germination—DOG) at 16 °C, with a photoperiod of 16/8 h day/night and a photon flux density of 250–280 μmol m$^{-2}$ s$^{-1}$ PAR. The humidity was 69%.

### 2.3. Morphological Analysis

　　　The seeds were kept in the growth chambers for 21 days, and daily measurements of the values were conducted to determine the index of germination and the index of germination velocity, according to the equations proposed by Maguire [26]. The weight of the biomass of 50 seedlings was measured at the end of this period.

### 2.4. Isolation and Identification of Fungi Colonizing Seeds

　　　Observations of the germination boxes (100 seeds for each combination) were performed every 24 h to record the development of fungal colonies on the seeds. The fungal cultures growing on the seeds were passed onto potato dextrose agar (PDA) (Biocorp LTD, Warsaw, Poland) medium and water agar. The plates were incubated at a temperature of 23 °C for 5–7 days under a 12 h lighting cycle. After the pure cultures had been isolated, macroscopic and microscopic observations were carried out to determine the species and the types of fungi. Pure fungi cultures were identified using the classical method. The following morphological traits were macroscopically determined: color, structure, height,

density of the aerial mycelium and the reverse, colony shape. A Nikon Eclipse E-200 MV (Tokyo, Japan) optical microscope with 200× magnification and computer image analysis were used to observe and evaluate the structure of vegetative and conidiogenous hyphae and the spores (asexual and sexual, their color, shape and size). The species of fungi were identified on the basis of mycological keys and monographs [27–32]. Identification of each isolate was conducted twice.

Grown colonies were checked under microscope, then colonies resembling *Streptomyces* were transferred to yeast malt extract (YME) agar plates and incubated at 30 °C for 15–20 days. Then single colonies were transferred to YME agar plates again to obtain pure cultures. Spore chain morphology was examined by light microscopy. Aerial spore mass colors were determined by morphological examination on YME [33].

The microorganisms that colonized the seeds were characterized using the following population parameters: frequency of occurrence of individual species C = a/b × 100 where: a—number of occurrences of the given species, and b—total number of isolates (100%). On this basis, microorganisms were assigned to frequency groups: eudominants—over 10.0% of the total isolated microorganisms, dominants 5.1–10%, subdominants 2.1–5%, recedents 1.1–2%, subrecedents below 1%.

### 2.5. Statistical Analysis

Statistical software Statistica version 13 was used to analyze the data by applying the analysis of variance (ANOVA) at a probability of $p < 0.01$; significant differences among means were defined using the Tukey's test. The letters above the error bars indicate the significance at $p < 0.01$ confidence level. Error bars represent the standard deviation (SD) values.

## 3. Results

### 3.1. Seeds Germination and Seedling Biomass

It was found that of all the examined cases, seeds germinated best after being photostimulated with a red (LR) or blue (LB) laser (Figure 1). The highest number of germinating seeds was found in the case of irradiation of the seeds plus vaccine.

The germination index did not vary significantly between the irradiated objects. The highest germination index (between 83–97% on the DOG 11) was noted in the case of seed irradiation plus vaccine, and the best values were found after treatment with LR. Slightly lower indexes were found in the case of vaccine irradiation (53–76% for DOG 11). The lowest germination index was found in the case of seed irradiation.

The experiment showed that the greatest biomass (0.435 g) was found in the case of vaccine irradiation with blue laser (Figure 2A). Compared to the other cases, more biomass was also found after seed were irradiated with a red laser and after the simultaneous use of red and blue laser.

Slightly higher seedling biomass was found when the seeds and vaccine were stimulated by red light laser compared to control (Figure 2B). In addition, after the use of the blue laser, significantly greater seedling biomass was found in the case of vaccine irradiation compared to the control.

### 3.2. Isolation and Identification of Fungi Colonizing Seeds

Table 1 presents all of the microorganisms that were isolated from the tested seeds. Twelve genera of microscopic fungi and actinomycetes of the genus *Streptomyces* were isolated from the analyzed soybean seeds. There were eight species of fungi that were classified as pathogenic to soybeans seeds. It was found that pathogenic species were more numerous in the fungus community. Species *B. cinerea* and *A. flavus* were found in the group of eudominants. The seeds were also colonized by four species of non-pathogenic fungi. Species *R. nigricans* was found in the dominant community while the species *G. roseum* was found in the subdominants group. It is effective in protecting germinating soybeans and the roots of older plants against infestation by soil-borne pathogenic fungi.

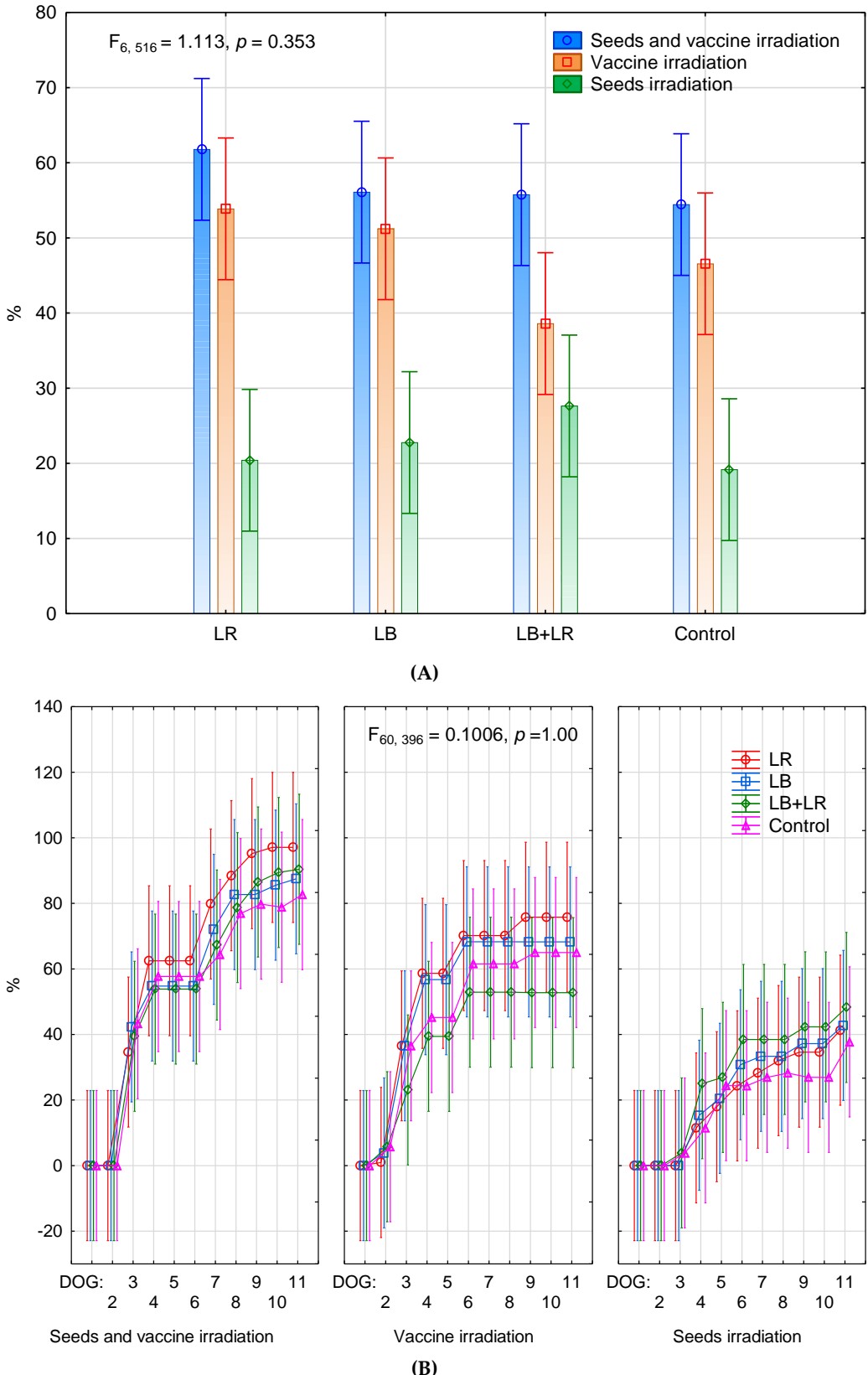

**Figure 1.** (**A**) Germination index (%) of soybean seeds with regard to irradiation treatments. (**B**) Germination index (%) of soybean seeds with regard to the second factor, i.e., the irradiated material and days of germination (DOG). LR—Helium-Neon laser (red light), LB—Argon laser (blue light), LB + LR—use of Ar laser followed by He-Ne laser.

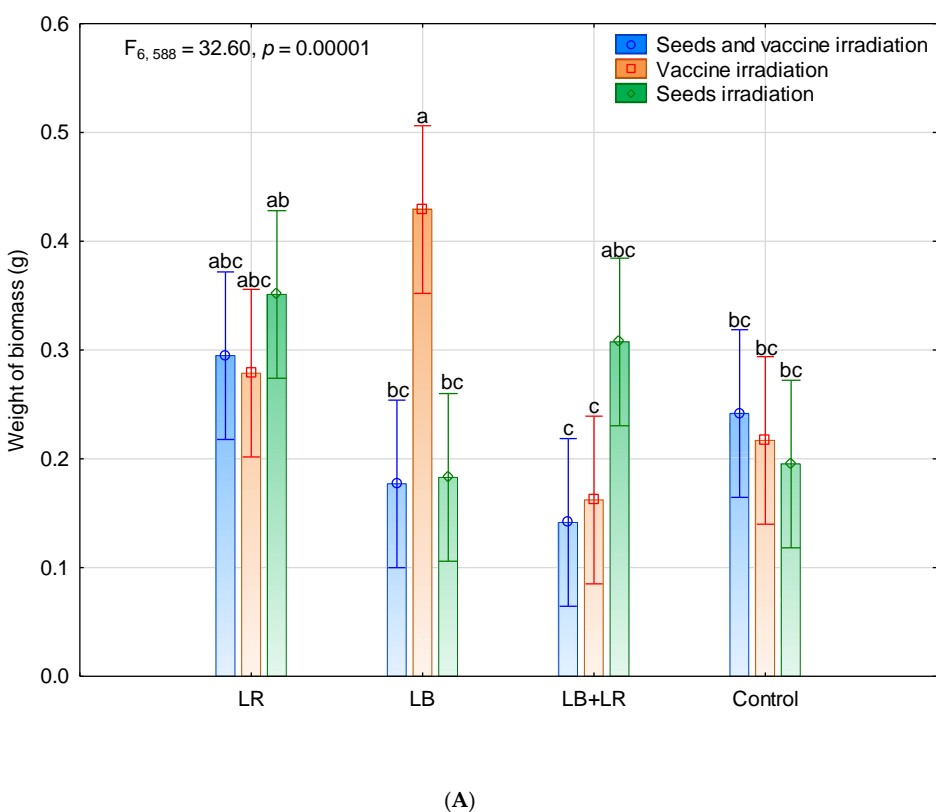

(**A**)

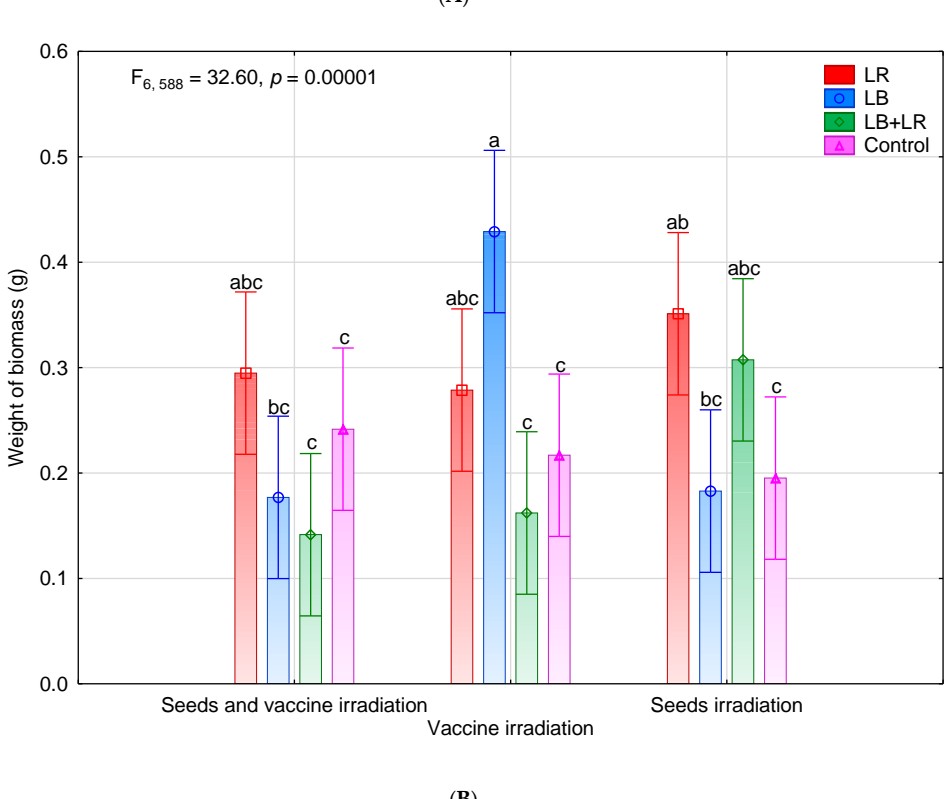

(**B**)

**Figure 2.** (**A**) Weight of biomass (g) of soybean seedling with regard to irradiation treatments, according to the first factor, i.e., the type of laser irradiation. Different letters indicate the significant differences at *p* < 0.01 by Tukey's test. (**B**) Weight of biomass (g) of soybean seedling with regard to irradiation treatments, according to the second factor, i.e., the irradiated material. Different letters indicate the significant differences at *p* < 0.01 by Tukey's test. LR—Helium-Neon laser (red light), LB—Argon laser (blue light), LB + LR—use of Ar laser followed by He-Ne laser.

**Table 1.** Frequency (%) of microorganisms isolated from the seeds.

| Microorganism Species | Seed and Vaccine Irradiation | Vaccine Irradiation | Seed Irradiation | Total | Group of Attendance |
|---|---|---|---|---|---|
| *Streptomyces* spp. | 0.00 | 0.00 | 25.00 | 25.00 | eudominants |
| **Pathogenic fungi** | | | | | |
| *Botrytis cinerea* | 0.00 | 29.29 | 0.00 | 29.29 | eudominants |
| *Aspergillus flavus* | 17.14 | 2.14 | 0.00 | 19.29 | eudominants |
| *Phoma glomerata* | 6.43 | 0.00 | 0.00 | 6.43 | dominants |
| *Rhizoctonia solani* | 0.00 | 1.43 | 0.00 | 1.43 | recedents |
| *Fusarium culmorum* | 0.71 | 0.00 | 0.00 | 0.71 | subrecedents |
| *Phomopsis sojae* | 0.71 | 0.00 | 0.00 | 0.71 | subrecedents |
| *Alternaria alternata* | 0.00 | 0.71 | 0.00 | 0.71 | subrecedents |
| *Colletotrichum* spp. | 0.00 | 0.71 | 0.00 | 0.71 | subrecedents |
| **Nonpathogenic fungi** | | | | | |
| Antagonistic fungi | | | | | |
| *Gliocladium roseum* | 2.14 | 0.00 | 0.00 | 2.14 | subdominants |
| Saprotrophic fungi | | | | | |
| *Rhizopus nigricans* | 0.00 | 7.86 | 1.43 | 9.29 | dominants |
| *Penicillium* spp. | 0.00 | 3.57 | 0.00 | 3.57 | subdominants |
| *Trichothecium roseum* | 0.71 | 0.00 | 0.00 | 0.71 | subrecedents |
| Total | 27.8 | 45.7 | 26.4 | 100 | |

Laser irradiation affected the communities of microorganisms isolated from seeds (Figure 3A–C). Six species of fungi were isolated from the case of seed and vaccine irradiation (Figure 3A). Species *A. flavus* constituted over 60% of the community. There was also a large number of *P. glomerate* and the antagonistic species of *G. roseum* constituted 8% of isolates. In contrast, *B. cinerea* was the most abundant species after vaccine irradiation (Figure 3B). *R. nigricans* species were the next most abundant species isolated in this case. The largest diversity in the fungal community was found where the vaccine was irradiated with lasers, and included two types and five species of fungi. In contrast, where the soybeans seeds were exposed to laser irradiation, the fungi constituted only 5.41% of the isolates (Figure 3C), and the other microorganisms were *Streptomyces* actinomycetes.

It was found that laser exposure increased the number of microorganisms isolated from soybean seeds (Figure 4). The development of microorganisms was most strongly stimulated by LR used separately or in conjunction with the LB. The largest increase in the number of microorganisms was observed in the case where seeds were subject to red laser irradiation as well as in the case where both vaccines and seeds were irradiated with the combined lasers.

Figures 5–7 show the presence of the species following all three irradiation treatments and in the control.

In the case of the seeds and vaccine irradiation, more *P. glomerata* species were isolated after LB, LR and LR + LB irradiation (Figure 5). No species of *G. roseum,* which is considered an antagonist to soil-borne pathogenic fungi, were found among the isolated fungi after LB irradiation.

In the case where the bacterial vaccine was irradiated with LR or LB irradiation, more CFUs of *B. cinerea* species were isolated (Figure 6). After LB irradiation, less CFUs of *R. nigricans* species were isolated, while, LR + LB irradiation increased the number of *R. nigricans* colonies.

In the seed irradiation case, *Streptomyces* spp. bacteria were mainly present on the seeds and the number of these increased after LR or LR + LB irradiation (Figure 7).

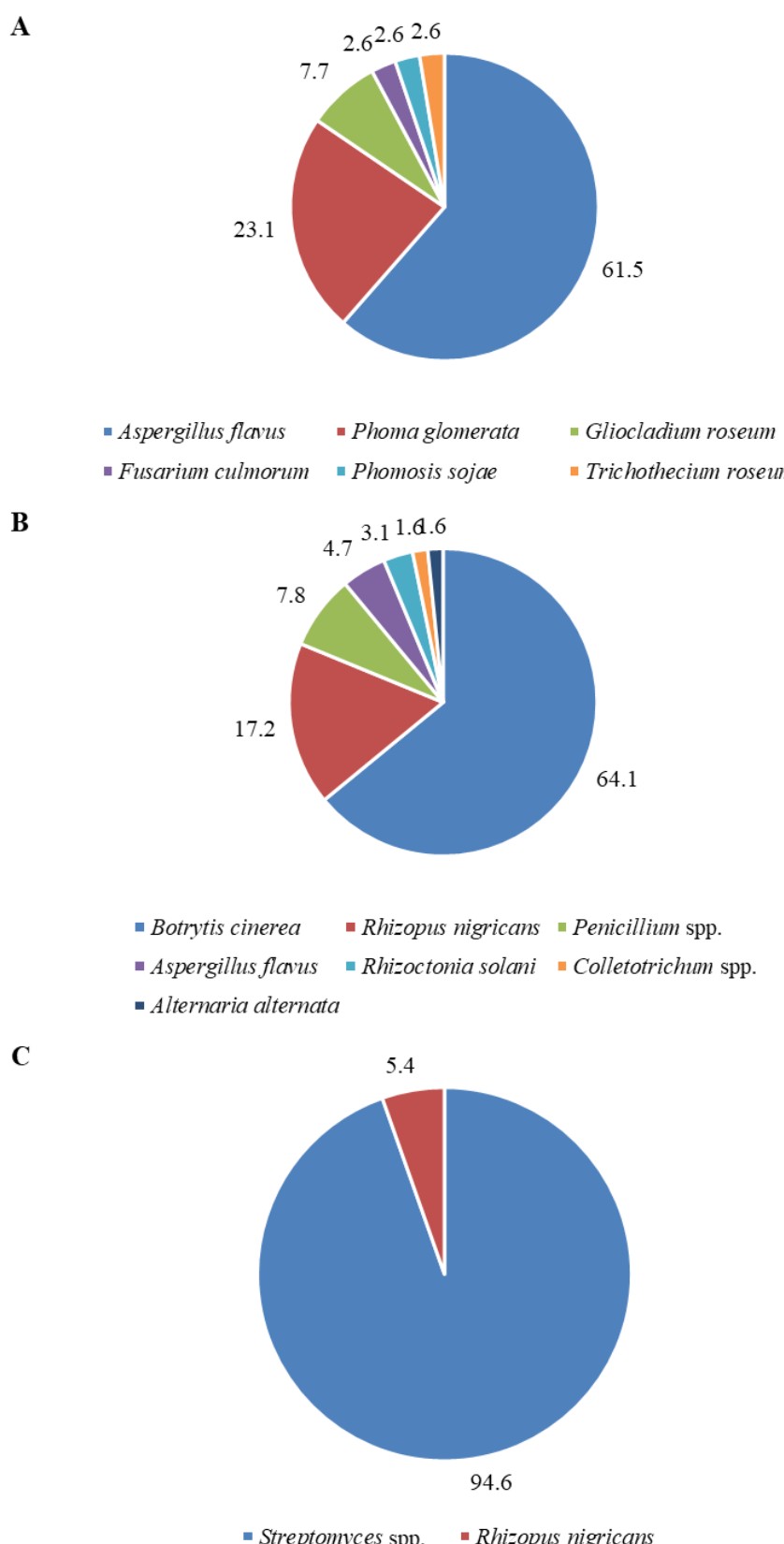

**Figure 3.** Share (%) of genera and species of fungi and *Streptomyces* spp. in microorganism communities colonizing seeds in the case of (**A**) seed and vaccine irradiation, (**B**) vaccine irradiation, and (**C**) seed irradiation.

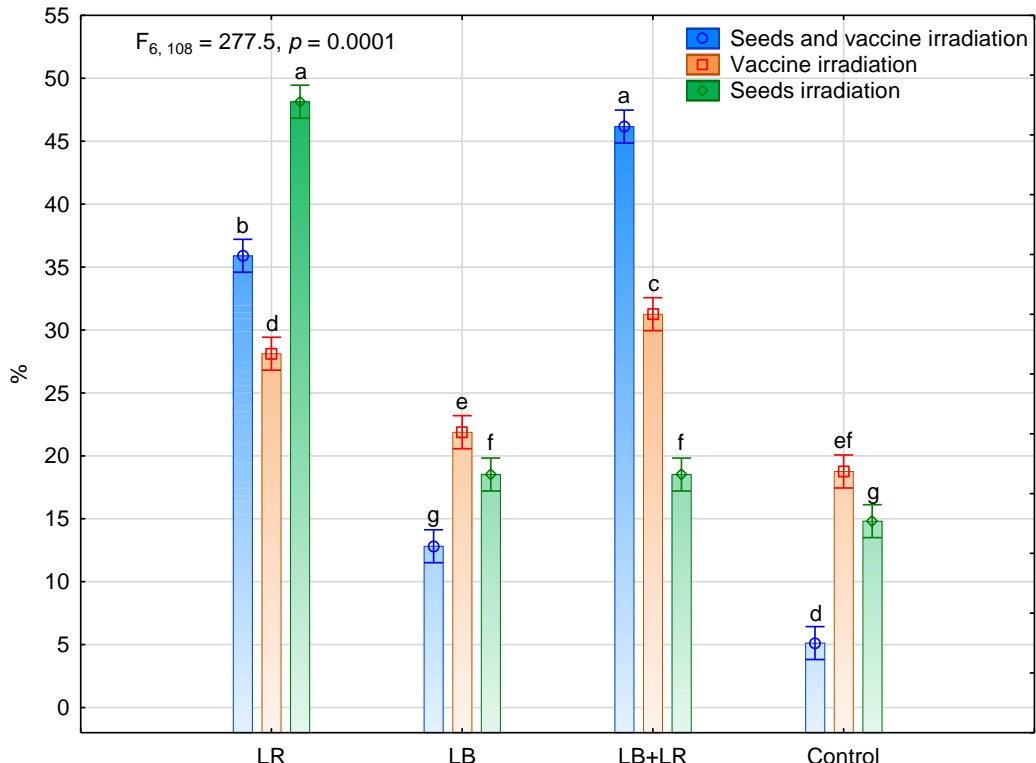

**Figure 4.** Frequency (%) of microorganisms' occurrence according to the type of laser irradiation. Different letters indicate the significant differences at $p < 0.01$ by Tukey's test. LR—Helium-Neon laser (red light), LB—Argon laser (blue light), LB + LR—use of Ar laser followed by He-Ne laser.

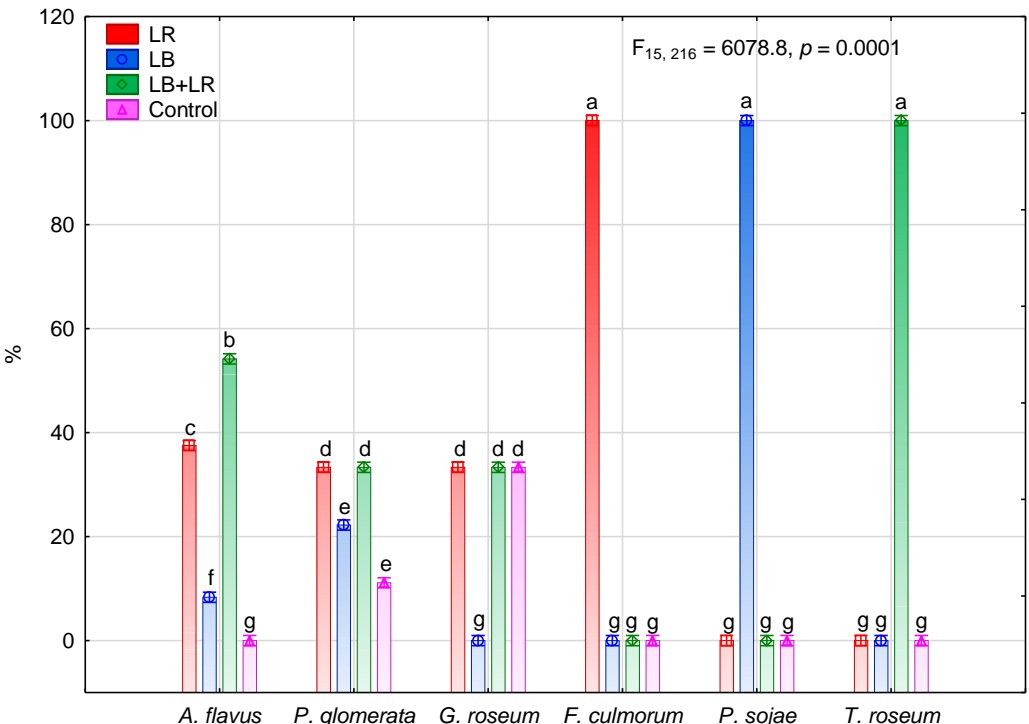

**Figure 5.** Frequency (%) of microorganisms' occurrence after seed and vaccine irradiation depending on the laser irradiation. Different letters indicate the significant differences at $p < 0.01$ by Tukey's test. LR—Helium-Neon laser (red light), LB—Argon laser (blue light), LB + LR—use of Ar laser followed by He-Ne laser.

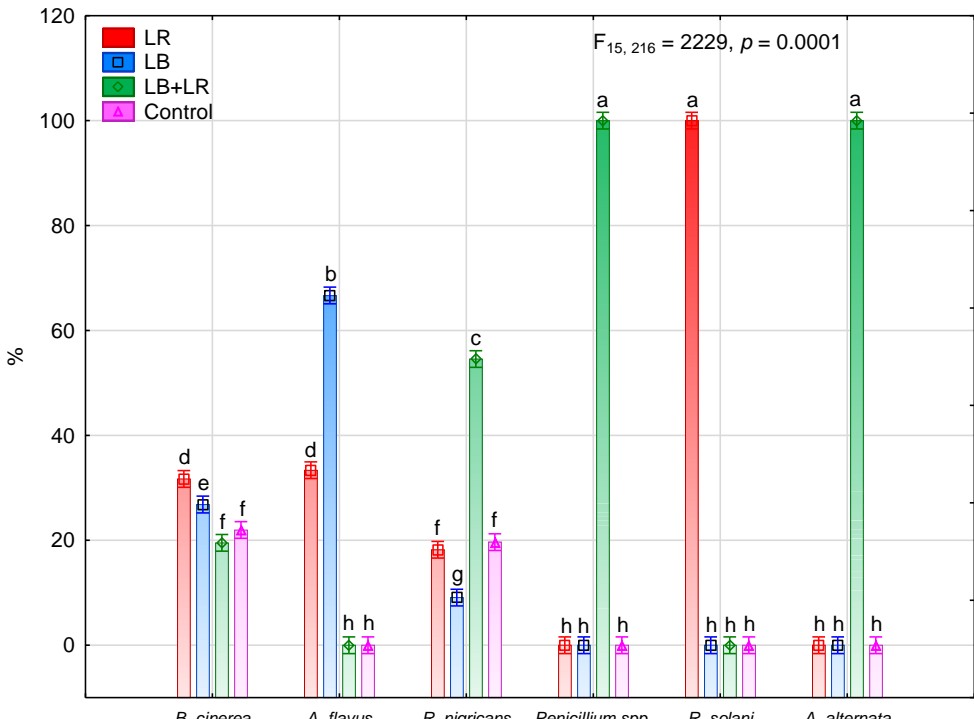

**Figure 6.** Frequency of microorganisms' occurrence after vaccine irradiation depending on the laser irradiation. Different letters indicate the significant differences at $p < 0.01$ by Tukey's test. LR—Helium-Neon laser (red light), LB—Argon laser (blue light), LB + LR—use of Ar laser followed by He-Ne laser.

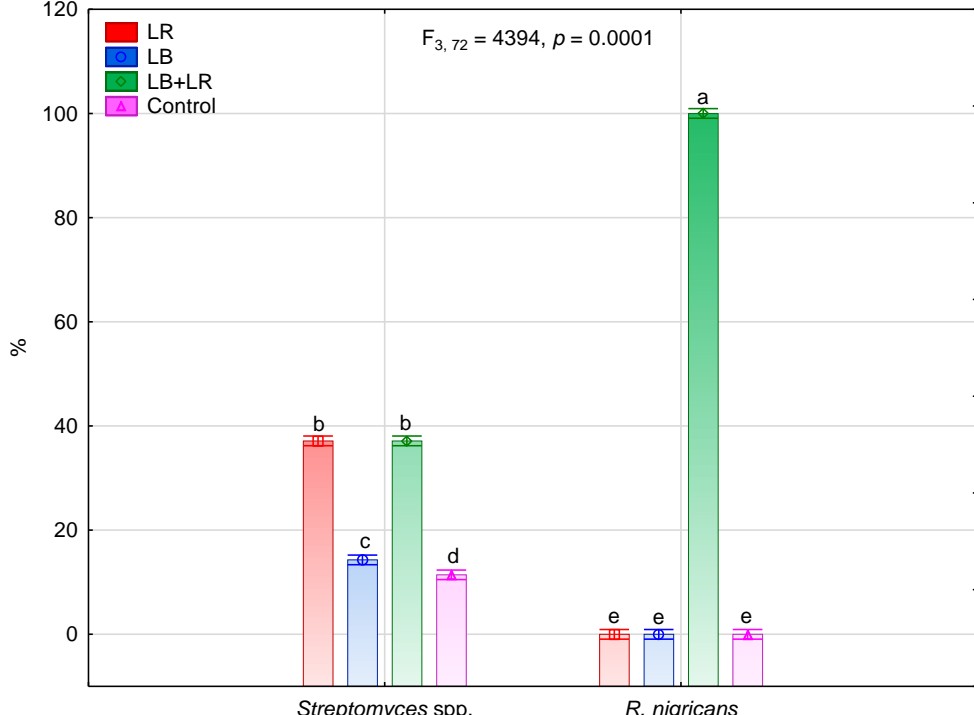

**Figure 7.** Frequency of microorganisms' occurrence after seed irradiation depending on the laser irradiation. Different letters indicate the significant differences at $p < 0.01$ by Tukey's test. LR—Helium-Neon laser (red light), LB—Argon laser (blue light), LB + LR—use of Ar laser followed by He-Ne laser.

## 4. Discussion

In the present study, the effects of pre-sowing laser stimulation on the germination and mycoflora abundance of soybean seeds were assessed. The germination capacity was between 17% and 61%. The laser light treatment resulted in an increase in the germination capacity and the weight of seedlings. Similar results were obtained by other authors in experiments done on other species [34–37]. Combined stimulation of the seeds and the vaccine had the least effect on seed germination, which was also related to a higher abundance of fungal diseases.

Twelve genera of microscopic fungi and actinomycetes of the genus *Streptomyces* were isolated from the analyzed soybean seeds. It was found that pathogenic species were more numerous in the fungus community. Species *B. cinerea* and *A. flavus* were most commonly isolated. Among non-pathogenic species, *R. nigricans* was found in the dominant community. In previous studies [12], 95 strains belonging to 40 species were isolated from soybean seeds. The dominant species were *Penicillium chrysogenum* and *Eurotium herbariorum*. Soybean mycocoenosis includes 74 types of fungi [38]. With regard to the number of taxa from separate mycofloras, 63 genera and about 108 or more species occur in seeds, 65 genera and about 88 or more species occur in pods, and 36 genera and approximately 47 or more species occur in flowers. Most of the fungi that occur in flowers can be cultured from pods, and the majority of those fungi occur in seeds [32]. Janda and Wolska [12] assessed the quantitative and qualitative composition of mold fungi inhabiting soybean seeds and found that the number of species in the samples ranged from 2–14. The species with an isolation frequency above 20% were: *Penicillium chrysogenum* (100%), *Eurotium herbariorum* (73.3%), *Aspergillus fumigatus* (46.7%), *Aspergillus versicolor* (40%), *Aspergillus sydowii* (26.7%) and *Rhizopus oryzae* (26.7%). In Kacaniova's [39] research on soya seeds, 26 species belonging to 13 genera of microscopic fungi were isolated and identified. Among the isolated genera, *Aspergillus* and *Penicillium*, were the most dangerous fungi, and occurred in 67–100% and 50–88% of the samples tested, respectively. Stored seeds can be seriously damaged by these genera due to the development of mold on grain as well as the production of mycotoxins and allergens. The major mycotoxin-producing fungi are species of *Aspergillus*, *Fusarium* and *Penicillium* and the important mycotoxins are aflatoxins, fumonisins, ochratoxins, cyclopiazonic acid, deoxynivalenol/nivalenol, patulin and zearalenone [40,41]. Species such as *Aspergillus fumigatus*, *Aspergillus flavus* etc. can contaminate and impair soybean used for feedstuff.

*Alternaria alternata*, *Fusarium oxysporum* and *Rhizoctonia solani* species were isolated from infected soybean seedlings whereas the main disease symptoms on soybean plants during the flowering phase were caused by *Fusarium oxysporum* f. sp. *glycines*. In addition, *A. alternata*, *F. culmorum*, *F. solani*, *R. solani* and *Sclerotinia sclerotiorum* were also isolated from diseased plants. Saprophytic fungi from the genera *Gliocladium*, *Penicillium* and *Trichoderma* were also isolated from soybean plants [42].

In our research, laser irradiation affected the abundance of *P. glomerata*, *B. cinerea*, *R. nigricans* and *G. roseum*. Laser irradiation usually increased the number of these fungi. The use of LB reduced the number of the non-pathogenic species *R. nigricans* and *G. roseum*. The laser LR + LB irradiation, on the other hand, increased the frequency of *R. nigricans*.

Studies by other authors have also indicated that laser irradiation affects the microflora of the seeds.

Laser irradiation of soybean seeds for 3 min caused a clear reduction in the number of seed-borne fungi, which became more pronounced as the irradiation time was extended. *Rhizoctonia solani*, *Alternaria tenuissima*, *Cercospora kukuchii* and *Colletotrichurm truncatum* were completely eliminated when the seeds were pretreated with a dye and irradiated for 10 min [11]. In the case of hard wheat seeds irradiated with SHG Nd-YAG laser, no fungi infection was recorded [19]. Laser irradiation significantly diminished the quantity of maize seeds infected with *Fusarium* spp. fungi [3]. Irradiation with He-Ne laser light of alfalfa seed (American variety 'Legend') destroyed fungi from the *Penicillium* genus completely, whereas it significantly stimulated the growth of fungi from the *Alternaria* genus [18]. Laser beam irradiation of seeds of the Polish hybrid alfalfa var. Radius significantly influenced the development of fungi of the *Penicillium* and *Alternaria* type [43]. The infection of clover seeds by fungi strains of the *Phoma* and *Penicillium* type was eliminated by laser beam. Laser treatment should not be

applied in the case of massive seed infection with fungi of the *Alternaria* type since a significant increase in fungi reproduction after laser irradiation has been noted [44]. Differences in the effects of the laser on fungi can be due to differences in the colonization of seed tissues. The data from Cortina et al. [16] indicated that fungi from the genera *Fusarium*, *Phomoposis*, *Cercospora* and *Colletotrichum* colonized the seed's internal tissue, and were thus largely protected. Fungi from the genera *Rhizoctonia*, *Aspergillus*, *Mucor*, *Penicillium* and *Rhizopus*, did not colonize the seed tissue, or at least did so to a very limited extent. Spores attached superficially to the seed coat on some seeds were protected by pockets of air or other physical barriers.

## 5. Conclusions

The application of empirically selected algorithms of laser irradiation and a *Bradyrhzobium japonicum* vaccine on soybean seeds led to a significant increase in germination. The best germination rate resulted from seed photostimulation with a 632.8 nm laser light (83–97% on the DOG 11) plus vaccine. Irradiation of the vaccine with 514 nm laser light resulted in the greatest enhancement in soybean seedling biomass. Treatment of soybean seeds with laser light eliminated the incidence of 3 out of 8 pathogenic fungi tested (including *Aspergillus flavus*), but this did not occur with irradiation with laser plus vaccine, or vaccine only.

A significant increase in the frequency of occurrence of the saprotrophic fungi *Rhizopus nigricans* resulted from the irradiation of soybean seeds with a 514 nm laser light followed by 325 and 632.8 nm. A higher incidence of *Streptomyces* actinomycetes was found after a 632.8 nm laser irradiation of soy seeds. The results from this study suggest that laser irradiation and the application of the bacteria *Bradryzobium japonicum* on soybean seeds is a useful innovation in agrotechnical practices that incorporate sustainable development.

**Author Contributions:** Conceptualization, A.K.-K. and J.W.D.; Formal analysis, J.D., A.K.-K., A.Ś.; Funding acquisition, J.D.; Investigation, A.K.-K., J.D., A.Ś.; Writing—original draft, A.K.-K., J.D.; Writing—review & editing, A.K.-K. and J.D. All authors have read and agreed to the published version of the manuscript.

**Funding:** This research was funded by the Ministry of Education.

**Conflicts of Interest:** The authors declare no conflict of interest.

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
