# Peer review of "Impact of Coherent Laser Irradiation on Germination and Mycoflora of Soybean Seeds—Innovative and Prospective Seed Quality Management"

_agriculture, doi:10.3390/agriculture10080314_

Round 1
Reviewer 1 Report
Dear authors,
Although this research has its merit, there are some problems that need to be cleared. The following is what I'm concerned with:
Introduction
- You state that bacterial vaccine has significant effect on soybean germination and seed health (L 64-65) but there are no citations to corroborate this statement. Although this is a well-known fact, some background should be included to validate your choice to investigate the effect of irradiation on seed inoculated with B. japonicum.
Materials and Methods
- The experimental design lacks information - how many irradiated seeds were planted for investigating germination and seedling biomass?, how many seeds were used for microflora isolation?, how many measurements were made?, what kind of soil was used for seed planting?, was the soil sterilised?..
- If one of the goals was to investigate B. japonicum inoculation as a factor, why didn't you include a variant with no inoculation as a control?
Results
- You state that seeds germinated best when treated with LR or LB (L 124-126) and that most germinating seeds are found in vaccine irradiation treatment but these statements should be corroborated with statistical analysis - are there significant differences?
- Since you compare seedling biomass of different irradiation treatments of the same second factor (seed+vaccine, vaccine, seed irradiation) (L 131-134), it would be advisable to organise Figure 2 in that way as well, i.e. to group the columns according to the second factor (3 groups x 4 columns instead of 4 groups x 3 columns). The same should be done for Figure 1.
- In determining the abundance of mo you pooled all the frequency data together, why is that? What does that tell you? Please elaborate!
- In L 138-139 you comment that most mo were obtained from the symbiosis where only vaccine was irradiated, but you can't tell whether that particular seed had more mo before vaccine irradiation and application to begin with. Maybe consider omitting this comment as it gives no useful information.
- On the basis of results presented in Figure 3, you concluded that laser irradiation affected the communities of microorganisms isolated from the seeds (L 153). Please elaborate on this. In my opinion, such statement can only be made if you had the same starting point, i.e. the same mo communities in control which then underwent irradiation. This is clearly not the case according to Figures 5-7. This experimental design is not suitable for determining the effect of irradiation on all mo communities. In order to determine if the frequency of the microorganisms changes after irradiation treatment, seed should be sterilised, inoculated with the most frequent and most significant microorganisms and then irradiated.
- Figure 4 results are questionable because you are again comparing the results of mo isolation from different treatments but you don't have the same starting point, as mentioned earlier. Please elaborate on your choice of methodology.
- In text explaining Figures 5-7 (L 171-187) you are commenting on the effect of irradiation on mo species, i.e. on their CFUs which is questionable since you don't have unified species across treatments. Maybe it would be a good idea to only comment on species present in all three irradiation treatments and in control.
Discussion
- General information on mo importance (for eg. L 202-206, L 211-224) is maybe better suited for the introduction.
- Please consider discussing only the effect of irradiation on germination, seedling biomass and mo abundance, as, in my opinion, the experimental design is not suitable for making conclusions on the effect of irradiation on mo.
Conclusion
- Please consider omitting the conclusion in L 256-259 for reasons stated earlier.
Other
- L 33 - omit the literature citation in brackets
- L 162 Figure 4 - Seed ansd... = Seed and...
- Figure 2 post hoc - cb = bc
- L 194 - extra space after abundance?
Hope this was helpful! Good luck and best wishes!
Author Response
We would like to thank for review. We correct paper according to suggestion. Below we presented detail explanation of comments.
Introduction
- You state that bacterial vaccine has significant effect on soybean germination and seed health (L 64-65) but there are no citations to corroborate this statement. Although this is a well-known fact, some background should be included to validate your choice to investigate the effect of irradiation on seed inoculated with japonicum.
- Response: According to the suggestion we have inserted the citation to support this statement.
Materials and Methods
- The experimental design lacks information - how many irradiated seeds were planted for investigating germination and seedling biomass?, how many seeds were used for microflora isolation?, how many measurements were made?, what kind of soil was used for seed planting?, was the soil sterilised?..
Response: The information about the experimental design is now included in the methods chapter.
- If one of the goals was to investigate japonicuminoculation as a factor, why didn't you include a variant with no inoculation as a control?
- Response: Our goal was to assess the effect of laser stimulation on seeds and vaccine together and apart. The control was not irradiated.
Results
- You state that seeds germinated best when treated with LR or LB (L 124-126) and that most germinating seeds are found in vaccine irradiation treatment but these statements should be corroborated with statistical analysis - are there significant differences?
- Response: We made a statistical analysis.
- Since you compare seedling biomass of different irradiation treatments of the same second factor (seed+vaccine, vaccine, seed irradiation) (L 131-134), it would be advisable to organise Figure 2 in that way as well, i.e. to group the columns according to the second factor (3 groups x 4 columns instead of 4 groups x 3 columns). The same should be done for Figure 1.
- Response: We made a statistical analysis and according to suggestion the figures were amended (grouped the columns according to the second factor).
- In determining the abundance of mo you pooled all the frequency data together, why is that? What does that tell you? Please elaborate!
- Response: We made a correction of frequencies dedicated to Table 1.
- In L 138-139 you comment that most mo were obtained from the symbiosis where only vaccine was irradiated, but you can't tell whether that particular seed had more mo before vaccine irradiation and application to begin with. Maybe consider omitting this comment as it gives no useful information.
- Response: We deleted this sentence.
- On the basis of results presented in Figure 3, you concluded that laser irradiation affected the communities of microorganisms isolated from the seeds (L 153). Please elaborate on this. In my opinion, such statement can only be made if you had the same starting point, i.e. the same mo communities in control which then underwent irradiation. This is clearly not the case according to Figures 5-7. This experimental design is not suitable for determining the effect of irradiation on all mo communities. In order to determine if the frequency of the microorganisms changes after irradiation treatment, seed should be sterilised, inoculated with the most frequent and most significant microorganisms and then irradiated.
Response: The experiment pattern proposed by Reviewer 1 is very interesting and we will want to use it in subsequent experiments, however, from a scientific point of view we used the methodology used by many researchers. In our work, we presented the methodology based on the work of Wilczek et. al. 2004, 2005 a, b, Ouf and Abdel-Hady 1999 and Rassam et al. 2013. The use of soybean seeds naturally inhabited by fungi was intended to refer to real conditions.
- Figure 4 results are questionable because you are again comparing the results of mo isolation from different treatments but you don't have the same starting point, as mentioned earlier. Please elaborate on your choice of methodology.
- Response: We used the methodology know in literature.
- In text explaining Figures 5-7 (L 171-187) you are commenting on the effect of irradiation on mo species, i.e. on their CFUs which is questionable since you don't have unified species across treatments. Maybe it would be a good idea to only comment on species present in all three irradiation treatments and in control.
Discussion
- General information on mo importance (for eg. L 202-206, L 211-224) is maybe better suited for the introduction.
- Response: We corrected the introduction and discussion sections.
- Please consider discussing only the effect of irradiation on germination, seedling biomass and mo abundance, as, in my opinion, the experimental design is not suitable for making conclusions on the effect of irradiation on mo.
- Response: The discussion was checked and corrected.
Conclusion
- Please consider omitting the conclusion in L 256-259 for reasons stated earlier.
- Response: The conclusion was corrected. We deleted the sentence from L 256-259 and we added an additional conclusion.
Other
- L 33 - omit the literature citation in brackets
- L 162 Figure 4 - Seed ansd... = Seed and...
- Figure 2 post hoc - cb = bc
- L 194 - extra space after abundance?
Response: We corrected shortcomings.

Reviewer 2 Report
Is this article related to pest or micro-organisms? Please review your title, research, and literature used.
Abstract
Re-write the abstract section. Start with background information.
Make the sentences simple and clear to understand. “From the analyzed soybean, 20 12 species belonging to 12 genera of microscopic fungi and actinomycetes of the genus Streptomyces were 21 isolated. Botrytis cinerea was the most numerous species of mushrooms. Aspergillus flavus also 22 belonged to the isolated species”
Introduction
Please include literature on disease and laser irradiation.
Materials and Methods
detailed experimental design required. Please explain all the experimental procedures in detail.
How many times the whole experiment was repeated to validate the results?
Results
Figures 1, 2, 4, 5, 6, 7: How the data average was taken? What is the standard error?
Re-organize figure 3.
Author Response
We would like to thank for review. We correct paper according to suggestion. Below we presented detail explanation of comments.
Is this article related to pest or micro-organisms? Please review your title, research, and literature used.
Response: We changed the title and removed the ‘pest’ word.
Abstract
Re-write the abstract section. Start with background information. Make the sentences simple and clear to understand. “From the analyzed soybean, 20 12 species belonging to 12 genera of microscopic fungi and actinomycetes of the genus Streptomyces were 21 isolated. Botrytis cinerea was the most numerous species of mushrooms. Aspergillus flavus also 22 belonged to the isolated species”
Response: The abstract was re-written and corrected.
Introduction
Please include literature on disease and laser irradiation.
Response: The references regarding diseases and laser irradiation were inserted.
Materials and Methods
detailed experimental design required. Please explain all the experimental procedures in detail.
Response: The methodology was corrected and improved. The methodology is now presented in detail.
How many times the whole experiment was repeated to validate the results?
Response: The answer was added into the text.
Results
Figures 1, 2, 4, 5, 6, 7: How the data average was taken? What is the standard error?
Response: It is now explained in detail in methodology. We had 4 replications.
Re-organize figure 3.
Respose: The figure 3 was corrected.

Reviewer 3 Report
Biophysical LCR-activation of seeds in the visible region is a fairly relevant field of research, the success of which depends on precisely selected parameters of the light source, exposure, and others. Moreover, there is an opinion that "Epigenetic responses could be detected in seeds in the range between 789-960 nm".
I believe that the article contains relevant empirical data to deepen knowledge of the mechanism of activation of germination.
For small comments, see the attached file below.

Author Response
We would like to thank for review. Correction was made in text. Below you will find comments to suggestion.
Response:
We corrected units according to SI.
We measure the time of lasser irradiation using our device, which has already waiting for IP licence.
The length of germination was included in the methodology.
The outcome of Statistica is presented on figures.
We put DOI into references.

Round 2
Reviewer 2 Report
Review 2:
Describe the type of equipment: The time duration was controlled by equipment our production (patent in progress), which has programmed controller.
Each figure should be self-explanatory. Please add description and details of abbreviations used. For eg- Figure 1 and 2. What is LR? LB? LR+LB?
Re-organize figure 3. (Still the figure size of sub-figures a,b,c are not uniform. Also, the numbering in not consistent.
Figure - 5 and 6: The y axis title and details are not organized. Also, the Tukey test letter are not placed uniformly.
Most of the corrections are technical, please review all figures and organize/correct them.
Author Response
Thank you very much for comments.
According to your suggestion we corrected figures, descriptions of figures and information about device.
- We proved description regarding to device
- We add description and details of abbreviations used. For eg- Figure 1 and 2. What is LR? LB? LR+LB. Now the figures are self explanatory.
- We corrected figures